# Research on Fabrication Techniques and Focusing Characteristics of Metalens

Yuhui Zhang [1,2,3,*], Yuegang Fu [1,2,3,†], Chenhao Ma [1,2,3,†], Bowei Yang [1,†] and Yuanzhi Zhao [1,†]

1   School of Optoelectric Engineering, Changchun University of Science and Technology, Changchun 130022, China; fuyg@cust.edu.cn (Y.F.); mach@cust.edu.cn (C.M.); 2020200040@mail.cust.edu.cn (B.Y.); zyzzyzzyz0331@163.com (Y.Z.)
2   Key Laboratory of Optoelectronic Measurement and Optical Information Transmission Technology of Ministry of Education, Changchun University of Science and Technology, Changchun 130022, China
3   Key Laboratory of Advanced Optical System Design and Manufacturing Technology of the Universities of Jilin Province, Changchun 130022, China
*   Correspondence: 2022800005@cust.edu.cn
†   These authors contributed equally to this work.

**Abstract:** Metalenses have recently attracted increased attention due to their remarkable characteristics. The fabrication technology of metalenses has also become an important research direction. In this study, we propose a metalens structure based on Au–MgF$_2$–Au in infrared waveband. The preparation process of the metalens included magnetron sputtering, electron beam evaporation, and electron beam exposure. A dose test was performed during the exposure process, adjusting the exposure dose to minimize the proximity effect after exposure. Then, SEM was used to measure the processed metalens structure, and FDTD software was used to build a model based on the metalens, simulating and analyzing its focusing characteristics. The results show that the size deviation produced during the processing has little effect on the functionality of the metalens. The processed metalens can also focus different polarized light incidences at different spatial positions: The metalens can focus at 4.97 μm for x-polarized light and focus at 13.5 μm for y-polarized light. Additionally, the metalens has good focusing effects with different working wavelengths. We believe that the processing method of metalens proposed in this paper provides guidance for the preparation of subwavelength metasurface structures, and our findings are beneficial in developing new methods of near-infrared regime manipulation.

**Keywords:** metalens; preparation; infrared waveband; different polarization; focus

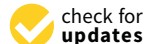



## 1. Introduction

Metasurfaces have characteristics that are not available in natural materials [1–9] and have obvious advantages in size, volume, processing [10,11], etc., meeting the requirements of miniaturization and integration in modern technology, making them key research objects for a large number of scientific researchers. Metalens is a two-dimensional plane lens based on metasurfaces, which realizes the focusing function of a traditional lens by integrating the transmission phase or reflection phase [12–14]. Metalens is different from the traditional lens; it has a subwavelength size, which is advantageous to the processing and miniaturization of the device. Based on the different wavelength and polarization states of the incident light, metalens can achieve a variety of different functions [15–24]. It has important applications in telescopes, cameras, and other imaging fields, as well as tunable devices, holographic displays, terahertz imaging, and nonlinear optics, and has great potential for development in the micro–nano field. Although many metalens structures have been proposed before, their processing is more difficult and has rarely been elaborated in previous studies; therefore, the fabrication technology of metalens and focusing characteristics of the processed metalens have not been fully investigated.

In this paper, we propose a metalens structure based on Au–MgF$_2$–Au in infrared waveband. The preparation process of the metalens includes magnetron sputtering (type, producer, city (state), country), electron beam evaporation, and electron beam exposure. A dose test was performed during the exposure process, adjusting the exposure dose to minimize the proximity effect after exposure. Then, a scanning electron microscope (SEM) was used to measure the processed metalens structure, and the finite-difference time-domain (Lumerical FDTD solutions 8.15.736.0) method was used to build a model based on the metalens, simulating and analyzing its focusing characteristics. The results showed that the size deviation produced during the processing has little impact on the functionality of the metalens. The processed metalens can focus on different polarized incidences at different spatial positions. For x-polarized incidence, the focal length is 4.97 μm. For y-polarized incidence, the focal length is 13.5 μm. For 45-degree linearly polarized incidence, there are two focal points, and the focal length are 4.97 μm and 13.5 μm, respectively. For circularly polarized incidence, there are also two focal points, and the focal point intensity is twice that of 45-degree linearly polarized light. The processed metalens also has good focusing effects with different working wavelengths. We believe that the processing method of metalens proposed in this paper provides guidance for the preparation of subwavelength metasurface structures, and our findings are beneficial in developing new methods of near-infrared regime manipulation.

## 2. Structural Design

Figure 1 illustrates the schematic of the proposed metalens, which is a metal–insulator–metal structure to form a Fabry–Perot cavity to enhance the interaction between light and resonance antenna. The material for the resonance antenna and the bottom mirror was selected as Au [21], while the material for the dielectric spacer was chosen as MgF$_2$ [22]. The thicknesses of Au antenna, MgF$_2$ spacer, and Au mirror were set as 30 nm, 50 nm, and 130 nm, respectively. The length and width of the unit cell of the metalens were both 200 nm. The x-direction of the metalens was composed of 60 unit structures, and the y-direction was periodically arranged for 30 cycles. The length and width of the resonance antenna can be calculated according to [14].

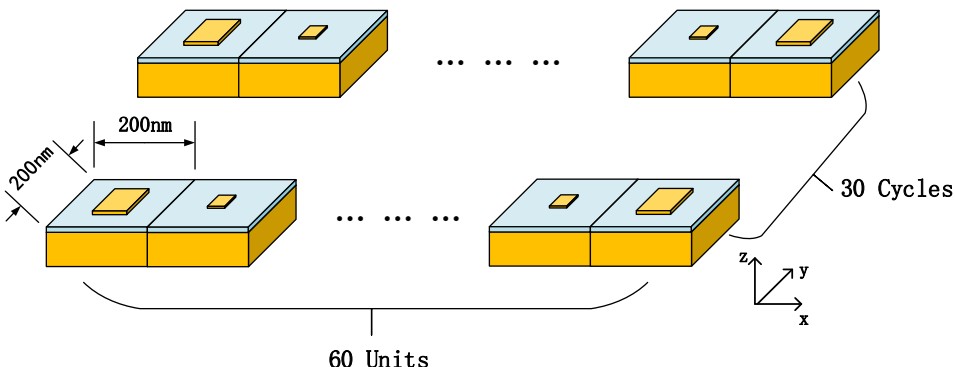

**Figure 1.** Diagram of the metalens structure.

The proposed structure can be designed to achieve the function of the focusing lens. To achieve desired focusing effect, the relative phase of the reflected wave at position x from the center should follow the expression [12].

$$\varphi(x) = \frac{2\pi}{\lambda_0} \left( \sqrt{(x + \Delta x)^2 + F^2} - F \right) \tag{1}$$

where $\lambda_0$ is the incident wavelength, $F$ represents the designed focal length, and $\Delta x$ represents the shift of focal point along the x-axis.

## 3. Preparation of the Structure

The preparation process of the metalens is shown in Figure 2. First, the silicon wafer was ultrasonically cleaned with acetone–isopropanol–water and dried with nitrogen, and then, a 130 nm Au film was grown on the silicon wafer by magnetron sputtering (Lab18, Kurt J. Lesker, PA, USA). In the second step, a $MgF_2$ film was deposited on the Au film by electron beam evaporation technology (Ei-5z, Ulvac, Kanagawa, Japan). The deposition rate and thickness of the $MgF_2$ film material were monitored by a quartz crystal; the film deposition rate was controlled at 0.2 nm/s, and the deposition thickness was 50 nm. In the third step, electron beam lithography (JBX5500ZA, JEOL, Tokyo, Japan) was used to prepare subwavelength Au nanoantennas. Here, the electron beam lithography technology mainly included the following key processes: spin coating, exposure, development, electron beam evaporation, stripping, etc. Spin coating photoresist on the surface of $MgF_2$ film was divided into two steps. The first step was to apply MMA, and the second step was to apply PMMA. A layer of Cr film was grown by magnetron sputtering before exposure. The reason for growing the 10 nm Cr film is to increase conductivity and reduce the proximity effect effectively. Then, the exposure and development processes began, during which the exposure dose varied from 100 to 1000 $\mu C/cm^2$. Before developing, it was necessary to use a Cr etching solution to remove Cr. The developer used a mixture of 4-Methyl-2-pentanone and isopropanol. After the above process was completed, the electron beam evaporation process was used to deposit Au. To improve adhesion between $MgF_2$ and Au, 5 nm of Ti was deposited as a binder, after which 25 nm Au was deposited. Finally, a peeling process was performed to obtain a metalens structure of Au–$MgF_2$–Au.

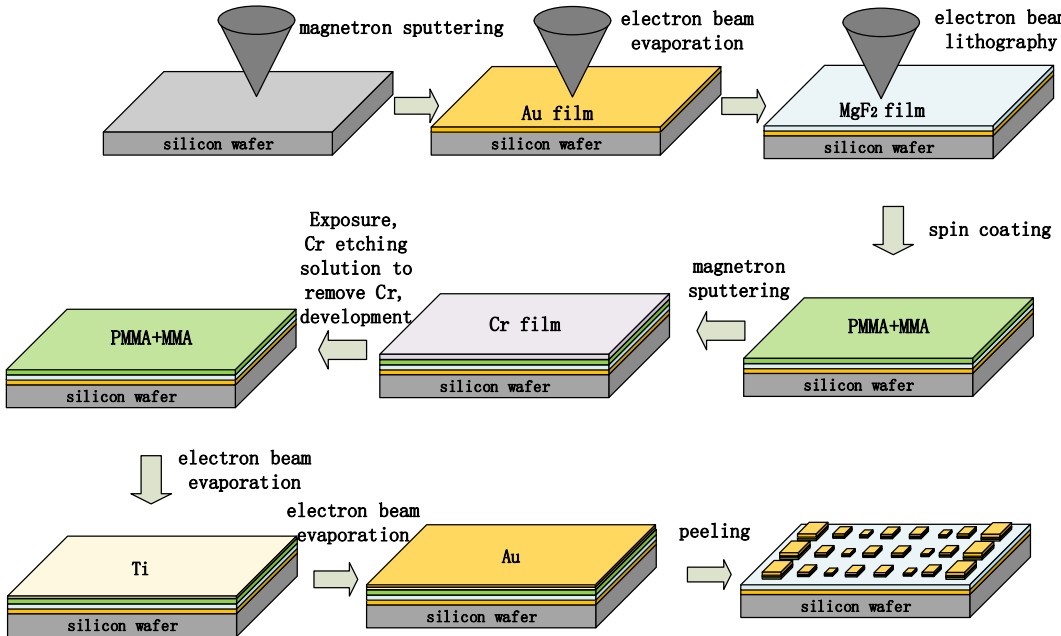

**Figure 2.** Process flowchart of the proposed metalens structure.

According to the above process, a batch of samples was processed, and the morphology was tested by the SEM (Eline Plus, Raith, Germany). The test results are shown in Figure 3. The results show that the effect of large size graphics is good, but the proximity effect of small size graphics is more significant. In electron beam lithography, due to the electron beam scattering in the glue layer and the substrate, outside the area where the glue layer is exposed, the adjacent areas are also exposed, resulting in the edge of the glue layer being blurred, and therefore, the shape of the graph expands to the adjacent areas, that is, the adjacent areas affected by the electron beam exposure. Therefore, it is necessary to dose and correct for the proximity effect before preparing samples. There are three different methods to correct the proximity effect of electron beam lithography: exposure dose correction,

figure size compensation, and background exposure compensation. Among them, dose correction is the most widely used method, and the most effective method. Therefore, we conducted a dose test during the exposure process; in other words, we adjusted the exposure dose to minimize the proximity effect after exposure.

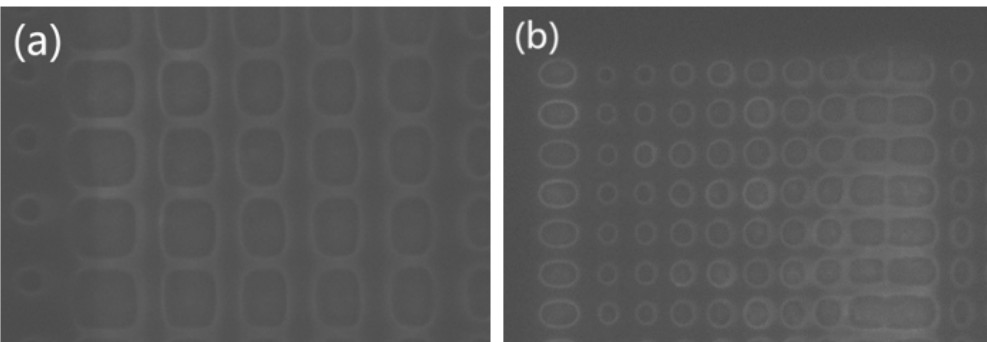

**Figure 3.** The scanning electron microscopic image (SEM) of samples: (**a**) the SEM image at 82,490× magnification; (**b**) the SEM image at 46,400× magnification.

Figure 4 illustrates the SEM testing of different exposure doses. Figure 4a is the SEM image with an exposure dose of 700 μC/cm$^2$; the middle part of the image is convex and irregular, indicating that the exposure dose is too large. When the exposure dose is 300 μC/cm$^2$, almost no image can be clearly presented, indicating that the dose is too small. Figure 4b is the SEM image with an exposure dose of 400 μC/cm$^2$; the proximity effect in the image is very small, but the morphology of small size graphics is not good. Figure 4c is the SEM image with an exposure dose of 500 μC/cm$^2$; the impact of proximity effect on patterns in the image is seen to be very small, and the morphology of small size graphics is relatively good. Figure 4d is the SEM image with an exposure dose of 600 μC/cm$^2$; the graphics appear deformed, indicating that the exposure dose is now too large. Therefore, an exposure dose of 500 μC/cm$^2$ was chosen to be used.

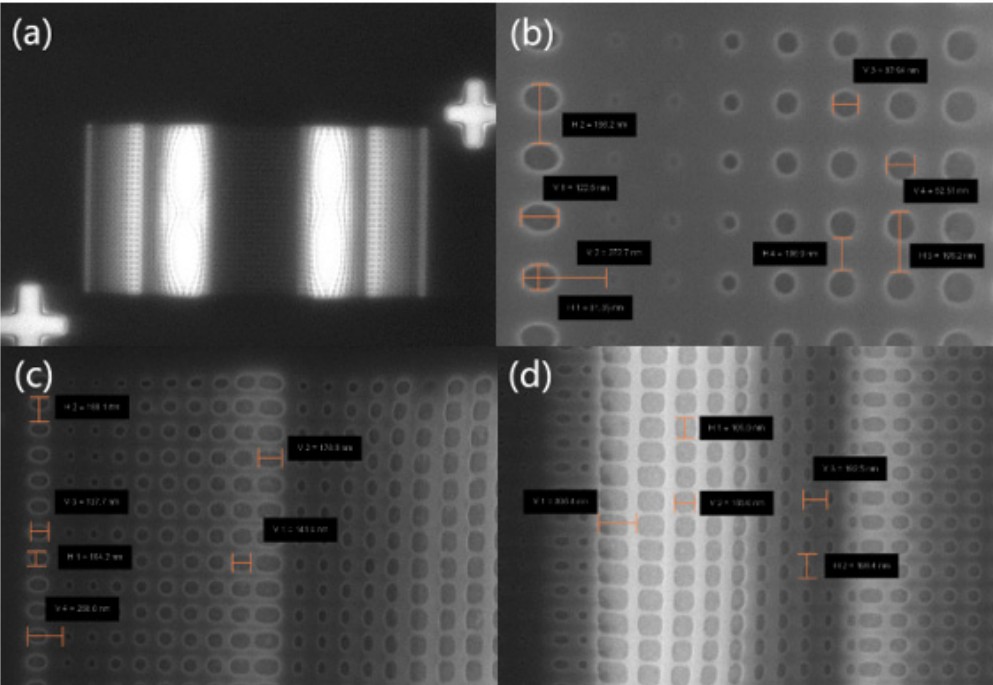

**Figure 4.** SEM images of different exposure doses: (**a**) the SEM image with an exposure dose of 700 μC/cm$^2$; (**b**) the SEM image with an exposure dose of 400 μC/cm$^2$; (**c**) the SEM image with an exposure dose of 500 μC/cm$^2$; (**d**) the SEM image with an exposure dose of 600 μC/cm$^2$.

## 4. Results and Discussion

SEM was used to measure the processed metalens structure, and FDTD software was used to build a model based on the processed metalens, simulating and analyzing its focusing characteristics. X-polarized light, y-polarized light, 45-degree linearly polarized light, and the circularly polarized light incidence with a wavelength of 800 nm were simulated, and the resulting reflected light fields are shown in Figure 5a,c,e,g, respectively. The intensity of the focusing spots along the x-direction is shown in Figure 5b,d,f,h. For x-polarized light, the focal length is 4.97 μm, and the full width at half maximum (FWHM) value of the focal spot is calculated to be 0.47 μm. For y-polarized light, the focal length is 13.5 μm, and the FWHM value of the focal spot is calculated to be 0.84 μm. For 45-degree linearly polarized light, there are two focal points, and the focal lengths are 4.97 μm and 13.5 μm, respectively. For circularly polarized light, there are also two focal points, and the focal lengths are 4.97 μm and 13.5 μm. It was also noted that the focal point intensity of circularly polarized light is twice that of 45-degree linearly polarized light. The results show that the processed metalens structure can focus at different positions when different polarized lights are incident.

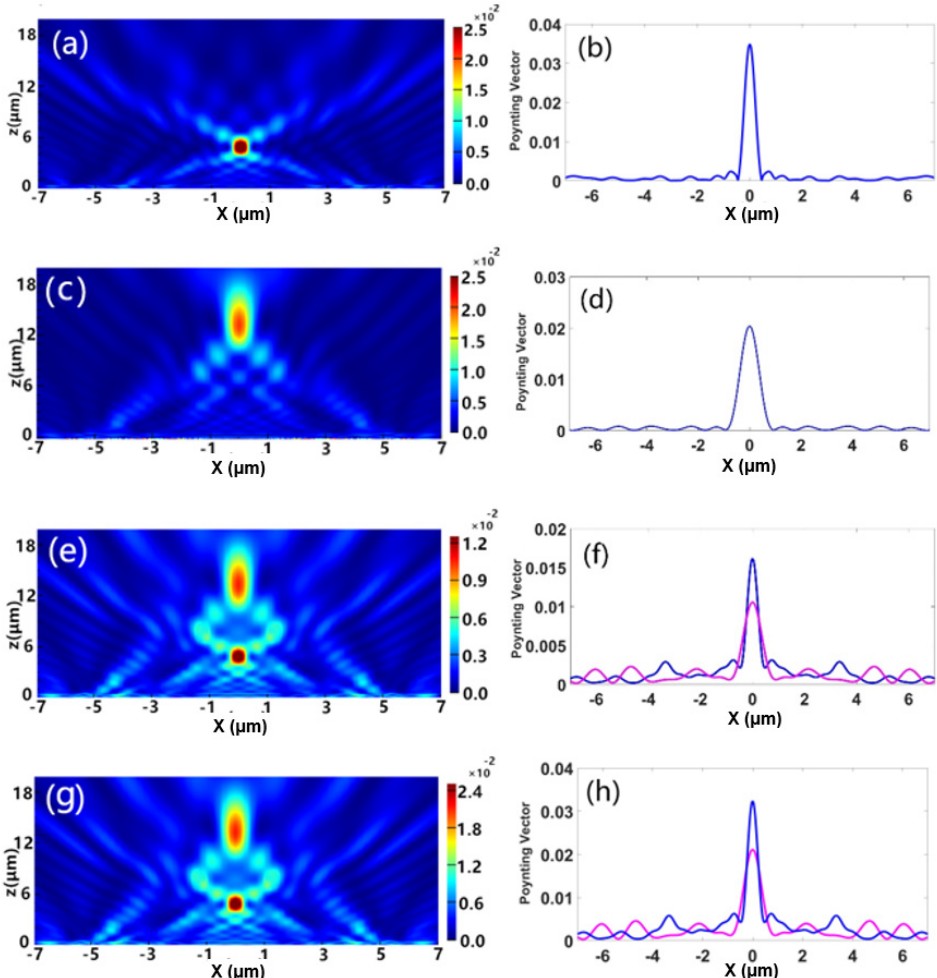

**Figure 5.** (**a**) The Poynting vector of the reflected light fields for the metalens with x- polarized incidence; (**b**) the intensities of the focusing spots along the x-direction for x- polarized incidence; (**c**) the Poynting vector of the reflected light fields for the metalens with y-polarized incidence; (**d**) the intensities of the focusing spots along the x-direction for y-polarized incidences; (**e**) the Poynting vector of the reflected light fields for the metalens with 45-degree linearly polarized incidence; (**f**) the intensities of the focusing spots along the x-direction for 45-degree linearly polarized incidence; (**g**) the Poynting vector of the reflected light fields for the metalens with circularly polarized incidence; (**h**) the intensities of the focusing spots along the x-direction for circularly polarized incidence.

Then, x-polarized incidences with different wavelengths (700 nm, 750 nm, 850 nm, 900 nm) were simulated, and the resulting reflected light fields are shown in Figure 6a–d. The results show that the processed metalens structure can also focus light incidences with different wavelengths.

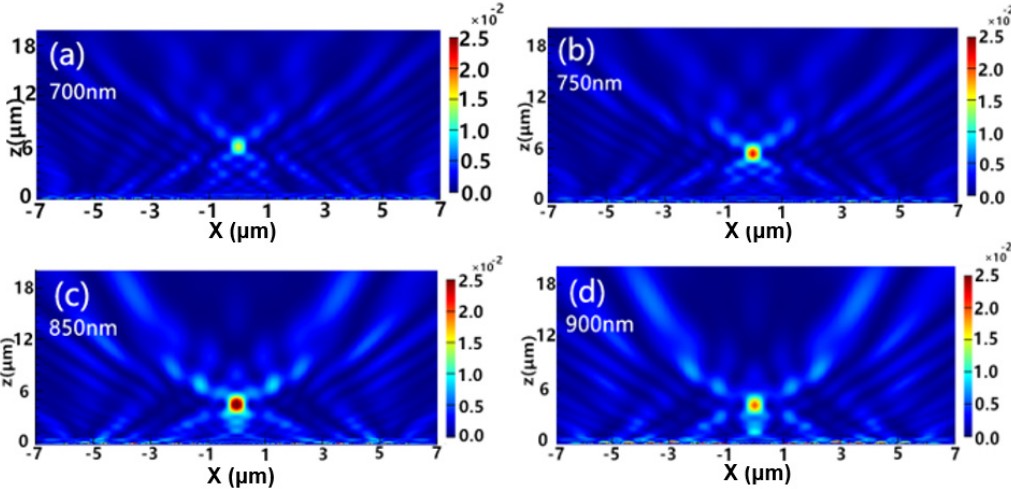

**Figure 6.** The Poynting vector of the reflected light fields for the metalens with incidence wavelength of (**a**) 700 nm, (**b**) 750 nm, (**c**) 850 nm, and (**d**) 900 nm.

Through modeling and testing of the experimentally prepared metalens structure samples, the results show that the size deviation produced during the processing has little effect on the functionality of the metalens. The processed metalens can also focus different polarized light incidences at different spatial positions and has good focusing effects with different working wavelengths. The processing method of metalens proposed in this paper provides guidance for the preparation of subwavelength metasurface structures.

The limitation of our research is the experimental conditions. We will complete a relevant experiment in the next step.

## 5. Conclusions

In summary, we proposed a metalens structure based on Au–MgF$_2$–Au in infrared waveband and tested it. SEM was used to measure the processed metalens structure, and FDTD software was used to build a model based on the metalens, simulating and analyzing its focusing characteristics. The results, both from simulation and experiment, show that the size deviation produced during the processing has little effect on the functionality of the metalens. The processed metalens can focus different, polarized light incidences at different spatial positions. For x-polarized incidence, the focal length is 4.97 μm. For y-polarized incidence, the focal length is 13.5 μm. For 45-degree linearly polarized incidence, there are two focal points, and the focal length are 4.97 μm and 13.5 μm, respectively. For circularly polarized incidence, there are also two focal points, and the focal point intensity is twice that of 45-degree linearly polarized light. The processed metalens also has good focusing effects with different working wavelengths. Compared with previous studies, the processed metalens in this study is more functional. We believe that the processing method of metalens proposed in this paper provides guidance for the preparation of subwavelength metasurface structures, and our findings are beneficial in developing new methods of near-infrared regime manipulation. Experimental characterization and detailed discussion on comparison with the calculation are essential. We will complete a relevant experiment in the next step.

**Author Contributions:** Conceptualization, Y.Z. (Yuhui Zhang) and Y.F.; methodology, Y.Z. (Yuhui Zhang); software, Y.Z. (Yuhui Zhang); validation, Y.Z. (Yuhui Zhang), Y.F. and C.M.; formal analysis, Y.Z. (Yuhui Zhang) and B.Y.; investigation, Y.Z. (Yuhui Zhang); resources, Y.Z. (Yuhui Zhang); data curation, Y.Z. (Yuhui Zhang); writing—original draft preparation, Y.Z. (Yuhui Zhang); writing—review and editing, Y.Z. (Yuhui Zhang), Y.F., C.M., B.Y. and Y.Z. (Yuanzhi Zhao); visualization, Y.Z. (Yuhui Zhang); supervision, Y.Z. (Yuhui Zhang); project administration, Y.Z. (Yuhui Zhang); funding acquisition, Y.F. and C.M. All authors have read and agreed to the published version of the manuscript.

**Funding:** This study received funding from "111" Project of China (D17017), Jilin Province Science and Technology Development Plan Project (20200403107SF), Jilin Province Science and Technology Development Plan Project (20210204191YY), and Changchun University of Science and Technology Youth Science Fund Project (XQNJJ-2017-08).

**Institutional Review Board Statement:** Not Applicable.

**Informed Consent Statement:** Not Applicable.

**Data Availability Statement:** Not applicable.

**Conflicts of Interest:** The authors declare no conflict of interest.

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
