# Peer review of "Research on Fabrication Techniques and Focusing Characteristics of Metalens"

_coatings, doi:10.3390/coatings12030359_

Round 1

Reviewer 1 Report

Dear authors,

the paper is interesting and proposes information for an actual research field; i.e. metalenses and metasurfaces. 

However, some comments and observations are addressed.

  1. Detail how was measured the thickness of fabricated thin films; i.e. Au antennas and the gold mirror.
  2. Detail what means the proximity effect.
  3. "When the exposure dose is 300 μC/cm2, almost no image can be clearly presented, indicating that the dose is too small." On what information is based this formation because in Figure 4 no reference related to this dose is given?
  4. "Then start the exposure and development processes, where the exposure dose was varied from 100 to 1000 μC/cm2." In the article is given information only for five exposure doses and no information about the step of the variation. Please, detail the results obtained for the other exposure doses that weren't mentioned in the article. 
  5. Explain what the FWHM acronym stands for and how was determined the mentioned values.
  6. If the 45-polarized incidence means that the angle of incidence was 45°, this should be mentioned in the article.
  7. Please explain why the focal length for x-polarized light, for 45 linearly polarized and circularly polarized light has the same value; i.e. 4.97 μm. 
  8. Please add the following information: the intensities of the focusing spots along the x-direction for the wavelengths 700 nm, 750 nm, 850 nm, and 900 nm. The results can be put together with those shown in Figure 6. Also, the focal lengths and the FWHM values for the wavelengths 700 nm, 750 nm, 850 nm, and 900 nm should be evaluated.
  9. Moderate English spelling is required. Please, ask a native speaker to help you. 
  10. Please use the same typing style in the whole manuscript; e.g. the Poynting vector is written either with capital or not. 

Author Response

Point 1: Detail how was measured the thickness of fabricated thin films; i.e. Au antennas and the gold mirror.

Response 1: Thanks for the advice. We use SEM to measure the cross section of the lens to directly measure the thickness of Au antennas and the gold mirror.

Point 2: Detail what means the proximity effect.

Response 2: Thanks for the advice. We have detailed what the proximity effect is in line 24 page 3 in the revised manuscript.

Point 3: "When the exposure dose is 300 μC/cm2, almost no image can be clearly presented, indicating that the dose is too small." On what information is based this formation because in Figure 4 no reference related to this dose is given?

Response 3: Thanks for the advice. Clear images were presented when the exposure doses were 700μC/cm2, 600μC/cm2, 500μC/cm2, 400μC/cm2. When the exposure measurement was 300μC/cm2, no clear image presentation indicated that the exposure dose was too small. Since there is no clear image, no relevant information of the exposure dose of 300μC/cm2 is added to Figure 4.

Point 4: "Then start the exposure and development processes, where the exposure dose was varied from 100 to 1000 μC/cm2." In the article is given information only for five exposure doses and no information about the step of the variation. Please, detail the results obtained for the other exposure doses that weren't mentioned in the article. 

Response 4: Thanks for the advice. Figure 4(a) is the SEM image with an exposure dose of 700μC/cm2, the middle part of the image is convex and irregular, indicating that the exposure dose is too large. Therefore, when the exposure dose is 700 to 1000μC/cm2, the exposure dose is too large. When the exposure dose is 300μC/cm2, almost no image can be clearly presented, indicating that the dose is too small. Therefore, when the exposure dose is 100 to 300μC/cm2, the exposure dose is too small. We only found the best value in the five exposure doses with 300μC/cm2, 400μC/cm2, 500μC/cm2, 600μC/cm2 and 700μC/cm2.

Point 5: Explain what the FWHM acronym stands for and how was determined the mentioned values.

Response 5: Thanks for the advice. We have explained the FWHM in line 2 page 5 in the revised manuscript, and the value that can be directly measured in the FDTD simulation results.

Point 6: If the 45-polarized incidence means that the angle of incidence was 45°, this should be mentioned in the article.

Response 6: Thanks for the advice. All the incident light in the paper is vertically. The 45-polarized incidence does not mean that the angle of incidence was 45°. As shown in the picture below, incidence angle 45° (a) means that the angle between the incident direction and its projection on the incident plane is 45°ï¼Œand 45-polarized incidence (b) means that the angle between the projection of the polarization direction on the incident plane and the X-axis is 45°.

Point 7: Please explain why the focal length for x-polarized light, for 45 linearly polarized and circularly polarized light has the same value; i.e. 4.97 μm. 

Response 7: Thanks for the advice. Both 45 linearly polarized and circularly polarized light can decompose into x-and y-polarized light, so that the focal length for x-polarized light, for 45 linearly polarized and circularly polarized light has the same value.

Point 8: Please add the following information: the intensities of the focusing spots along the x-direction for the wavelengths 700 nm, 750 nm, 850 nm, and 900 nm. The results can be put together with those shown in Figure 6. Also, the focal lengths and the FWHM values for the wavelengths 700 nm, 750 nm, 850 nm, and 900 nm should be evaluated.

Response 8: Thanks for the advice. Figure 6 only illustrates that the processed metalens can achieve focusing function when light of different wavelengths is incident, so we did not add the above information.

Point 9: Moderate English spelling is required. Please, ask a native speaker to help you. 

Response 9: Thanks for the advice. We have modified the English spelling in the revised manuscript.

Point 10: Please use the same typing style in the whole manuscript; e.g. the Poynting vector is written either with capital or not.

Response 10: Thanks for the advice. We have modified similar errors in the revised manuscript.

Reviewer 2 Report

This paper describes the experimental fabrication and numerical simulation of reflection-type cylindrical metalens working in the infrared region. Exposure dose optimization has been achieved to obtain the best size of the metalens.

However, although the lens was fabricated, this paper completely lacks experimental characterization of the fabricated lens. Although the FDTD simulation was performed based on the fabricated dimensions, it is too incremental from the authors' previous simulation paper (ref. 14, Y. Zhang et al., Coatings 2020, 10, 389) To meet to the criteria of Coatings, experimental characterization and detailed discussion on comparison with the calculation must be essential. So, the reviewer cannot agree to publish this manuscript in the current form.

Author Response

Response: Thanks so much for your comments. We fabricated the lens based on the previous simulation paper (ref. 14, Y. Zhang et al., Coatings 2020, 10, 389), experimental characterization and detailed discussion on comparison with the calculation are essential, but we temporarily do not have the experimental conditions. We will complete the relevant experiment in the next step, and your opinion is a guide to our next work.

Reviewer 3 Report

Nice presentation

Author Response

Response: We have revised the paper and thank you very much for your comments.

Reviewer 4 Report

The manuscript “Research on Fabrication Techniques and Focusing Characteristics of Metalens” (coatings-1590694) by Zhang et al. propose a metalens structure based on Au-MgF2-Au in infrared waveband. The topic is interesting, but I think this article should reconsider after proper changes in major revision for publication in Coatings.

  1. I would encourage and advise the authors to adopt some of the additional references published by MDPI in the introduction section:

Tresca Stress Simulation of Metal-on-Metal Total Hip Arthroplasty during Normal Walking Activity. Materials (Basel). 2021, 14, 7554. https://doi.org/10.3390/ma14247554

The Effect of Bottom Profile Dimples on the Femoral Head on Wear in Metal-on-Metal Total Hip Arthroplasty. J. Funct. Biomater. 2021, 12, 38. https://doi.org/10.3390/jfb12020038

  1. I see some errors on English in some areas of the present manuscript. To improve the quality of English used in this manuscript and make sure English language, grammar, punctuation, spelling, and overall style are correct, further proofreading is needed. As an alternative, the authors can use the MDPI English proofreading service for this issue.
  2. In the abstract section, the authors are encourage and advise to add quantitative results rather than not only qualitative results.
  3. The state of the art, the significance of the present study, and research novelty are not clearly present, the authors should highlight it more advanced in the introduction section.
  4. In the introduction section, the authors should explain the previous research conducted and its shortcomings. It will uphold the research gap that you filled with your research novelty. I recommend the authors elaborate their introduction section.
  5. Equation (1) should give the reference for supporting the expression.
  6. The author must provide a detailed specification and use condition more detail regarding all tools used in the research carried out so that the reader can estimate the accuracy and differences in the results that the authors describe due to the use of different tools in future studies.
  7. In the results and discussion section, authors are advised to compare the results they obtain with previous similar/identical studies if it is possible.
  8. In the last paragraph of the results and discussion section, the authors should add of one paragraph about the limitations of the research conducted.
  9. The conclusion of the present manuscript is not solid. Further elaboration is needed.
  10. Further research needs to be explained in the conclusion section.
  11. Please make sure the authors have used the Coatings, MDPI format correctly. The authors can download published manuscripts by Coatings, MDPI, and compare them with the present author's manuscript to ensure typesetting is appropriate. Some issues regarding typesetting are as follows:
  • The typesetting for put each authors email are not correct
  • Are of the font in the Keywords should be lowercase
  • For sentence with two or more references should be [1-3], not [1], [2], [3]
  • Author contributions, Institutional Review Board Statement, Data Availability Statement, Acknowledgments, and Conflicts of Interest are missing that should be added after the conclusion and before references.
  • Using et al. for the list of the authors in the references is only for 11th authors and beyond, if the number of authors is not more than 10, then all of their names must be included
  • And others

Author Response

Point 1: I would encourage and advise the authors to adopt some of the additional references published by MDPI in the introduction section:

Tresca Stress Simulation of Metal-on-Metal Total Hip Arthroplasty during Normal Walking Activity. Materials (Basel). 2021, 14, 7554. https://doi.org/10.3390/ma14247554

The Effect of Bottom Profile Dimples on the Femoral Head on Wear in Metal-on-Metal Total Hip Arthroplasty. J. Funct. Biomater. 2021, 12, 38. https://doi.org/10.3390/jfb12020038

Response 1: Thanks for the advice. We have adopt the additional references (references [21,22] in the revised manuscript) published by MDPI in the introduction section.

Point 2: I see some errors on English in some areas of the present manuscript. To improve the quality of English used in this manuscript and make sure English language, grammar, punctuation, spelling, and overall style are correct, further proofreading is needed. As an alternative, the authors can use the MDPI English proofreading service for this issue.

Response: Thanks for the advice. We have modified the English language, grammar, punctuation, spelling in the revised manuscript.

Point 3: In the abstract section, the authors are encourage and advise to add quantitative results rather than not only qualitative results.

Response 3: Thanks for the advice. We have added quantitative results in the abstract section (line 8-12) of the revised manuscript.

Point 4: The state of the art, the significance of the present study, and research novelty are not clearly present, the authors should highlight it more advanced in the introduction section.

Response 4: Thanks for the advice. We have highlighted the state of the art, the significance of the present study, and research novelty in the introduction section (line 12-15) of the revised manuscript.

Point 5: In the introduction section, the authors should explain the previous research conducted and its shortcomings. It will uphold the research gap that you filled with your research novelty. I recommend the authors elaborate their introduction section.

Response 5: Thanks for the advice. We have explain the previous research conducted and its shortcomings in the introduction section of the revised manuscript.

Point 6: Equation (1) should give the reference for supporting the expression.

Response 6: Thanks for the advice. We have given the reference [12] for supporting the expression.

Point 7: The author must provide a detailed specification and use condition more detail regarding all tools used in the research carried out so that the reader can estimate the accuracy and differences in the results that the authors describe due to the use of different tools in future studies.

Response 7: Thanks for the advice. All the results of the simulations in the article are provided by Lumerical FDTD solutions 8.15.736.0.

Point 8: In the results and discussion section, authors are advised to compare the results they obtain with previous similar/identical studies if it is possible.

Response 8: Thanks for the advice. We have compared the results with previous similar studies in the conclusion (line 12) of the revised manuscript.

Point 9: In the last paragraph of the results and discussion section, the authors should add of one paragraph about the limitations of the research conducted.

Response 9: Thanks for the advice. We have added one paragraph about the limitation of our research. We added the paragraph in line 9 page 6 in the revised manuscript.

Point 10: The conclusion of the present manuscript is not solid. Further elaboration is needed.

Response 10: Thanks for the advice. Further elaboration of the conclusion is necessary. And experimental characterization and detailed discussion on comparison with the calculation are essential, but we temporarily do not have the experimental conditions. We will complete the relevant experiment in the next step.

Point 11: Further research needs to be explained in the conclusion section.

Response 11: Thanks for the advice. We have explained our further research in the conclusion section (line 15-17) of the revised manuscript.

Point 12: Please make sure the authors have used the Coatings, MDPI format correctly. The authors can download published manuscripts by Coatings, MDPI, and compare them with the present author's manuscript to ensure typesetting is appropriate. Some issues regarding typesetting are as follows:

  • The typesetting for put each authors email are not correct
  • Are of the font in the Keywords should be lowercase
  • For sentence with two or more references should be [1-3], not [1], [2], [3]
  • Author contributions, Institutional Review Board Statement, Data Availability Statement, Acknowledgments, and Conflicts of Interest are missing that should be added after the conclusion and before references.
  • Using et al. for the list of the authors in the references is only for 11th authors and beyond, if the number of authors is not more than 10, then all of their names must be included
  • And others

Response 12: Thanks for your corrections. All of the above questions have been modified.

Round 2

Reviewer 1 Report

Dear authors,

the manuscript was improved after revision. In this form, it can be published. 

All the best!

Reviewer 2 Report

The authors didn't respond to me. 

Reviewer 4 Report

Dear Yuhui et al.,

After carefully reading the author's revised manuscript entitled "Research on Fabrication Techniques and Focusing Characteristics of Metalens" (Manuscript ID: coatings-1590694) by Yuhui et al., The authors have been made significant improvements in the revised manuscript. Also, all of the issue in my review report has been addressed precisely.

With my pleasure, I recommend the manuscript should be accepted for publication on Coatings.

Best regards,

The Reviewer